# From Endophyte Community Analysis to Field Application: Control of Apple Canker (*Neonectria ditissima*) with *Epicoccum nigrum* B14-1

**Matevz Papp-Rupar \*, Leone Olivieri, Robert Saville, Thomas Passey**  **, Jennifer Kingsnorth, Georgina Fagg, Hamish McLean**  **and Xiangming Xu** 

NIAB, New Road, East Malling, Kent ME19 6BJ, UK
* Correspondence: matevz.papp-rupar@niab.com

**Abstract:** Apple canker, caused by *Neonectria ditissima* (Tul. and C. Tul.) Samuels and Rossman, is a major disease of apples (*Malus domestica*) worldwide. *N. ditissima* infects through natural and artificial wounds. Infected wood develops canker lesions which girdle branches and main stems causing reduced yield and tree death. *N. ditissima* is difficult to control; removal of inoculum (cankers) is expensive and therefore seldom practiced, whilst effective chemical products are being banned and no biocontrol products have been found to be effective against *N. ditissima*. This study used cues from a previous apple endophyte community analysis to isolate and test fungal endophytes belonging to the genus *Epicoccum* as potential endophytic biocontrol agents. *Epicoccum nigrum* B14-1, isolated from healthy apple trees, antagonised *N. ditissima* in vitro and reduced the incidence of *N. ditissima* infections of leaf scars by 46.6% and pruning wounds by 5.3% in field conditions at leaf fall. Autumn application of B14-1 conidia increased *E. nigrum* abundance in apple tissues at 10–20 days post-inoculation by ca. 1.5×, but this returned to control levels after one year. *E. nigrum* B14-1 did not cause detrimental effects on apple foliage, buds, fruit, or growth and could therefore present a new biocontrol agent to manage *N. ditissima* in commercial apple production.

**Keywords:** *Neonectria ditissima*; *Epicoccum nigrum*; endophyte; biocontrol; field study; colonisation; host fitness



## 1. Introduction

European apple canker, caused by *Neonectria ditissima* (Tul. and C. Tul.) Samuels and Rossman, is one of the most destructive diseases of apple (*Malus domestica*) worldwide [1]. *N. ditissima* infects wood through natural and artificial wounds such as pruning cuts [2], leaf scar wounds [3], bud scale scars, and fruit-picking wounds [4]. While infectious conidia and ascospores are produced by *N. ditissima* cankers all year around [5,6], leaf scar infections at leaf fall are considered the main entry points in European apple-growing regions. As few as 5–10 spores are sufficient to establish an infection [7,8], upon which lesions can rapidly girdle branches and main stems causing yield decline in older orchards and up to 30% tree mortality in newly established orchards [9].

Effective *N. ditissima* control is difficult to achieve. Most commercial apple cultivars are highly susceptible to *N. ditissima*. Breeding resistant varieties may be possible, but it is a long process given the polygenic nature of resistance against *N. ditissima* [10,11]. Removal of inoculum (infected tissue) is effective but also very expensive, time and labour-intensive and therefore seldom implemented in commercial apple production. Chemical protection is hampered by recent dose reductions and bans of effective plant protection products such as carbendazim and copper oxychloride [4,6]. Rainfall during high wound availability periods (such as picking and leaf fall) can further increase disease risk by increasing *N. ditissima* sporulation and dispersal. So far, no commercially available biocontrol agents have been found to be effective against *N. ditissima* in field conditions [12,13]. Recent research used a

systematic approach to screen over 500 fungal strains stepwise to identify four *Clonostachys rosea* isolates with high efficacy in detached branch tests and with commercially relevant cold, drought, and UV tolerance [14]. Their efficacy in field conditions, however, has not been tested.

The presence of particular endophytes and differences in endophyte community composition can affect plant properties more than genetic variation intrinsic to the host [15,16]. Endophytes have been shown to contribute to host disease resistance [17–20] and also other agronomic traits such as growth promotion and abiotic stress tolerance [21]. Recent reports suggest that apple endophyte communities vary due to host genotype and susceptibility to European apple canker [22–24]. It is therefore possible that endophytic communities play a role in host susceptibility. According to Schulz and Boyle [25], the majority of aboveground fungal endophytes in trees are non-systemic with a broad host range (Class III endophytes), implying that a single strain could colonise a range of host species and cultivars. Endophytes could be especially effective the in control of *N. ditissima* due to their capacity to colonise apple tissues prior to wound formation and thus prevent pathogen entry.

In a study conducted in 2016, internal transcribed spacer (ITS) amplicon sequencing was used to investigate fungal endophytic communities in one-year-old apple shoots of cultivars highly susceptible (Gala, Braeburn) and relatively resistant (Grenadier, Golden Delicious) to *N. ditissima*. Overall fungal community structure varied between canker-susceptible and resistant cultivars [6], with several fungal operational taxonomic units (OTUs) being significantly more abundant in resistant cultivars. These taxonomic groups could directly or indirectly contribute to the observed differences in cultivar susceptibility to *N. ditissima*. The most promising OTU belonged to the *Epicoccum* genus. It was highly abundant in all samples and its relative abundance in resistant cultivars was approximately 10 times higher than in susceptible cultivars. Moreover, isolates belonging to *Epicoccum nigrum* have previously been shown to antagonise plant pathogens [26–30]. In this study we followed up on these cues from the sequencing study [6]. Our two hypotheses were: (a) higher abundance of *Epicoccum* strains in canker-resistant cultivars indicates their biocontrol potential against *N. ditissima*, and (b) due to their high abundance in healthy trees across cultivars, they are likely to be harmless and effective colonisers of the apple endosphere.

Targeted endophyte screening was used to isolate endophytic *E. nigrum* strains from asymptomatic apple shoots. A range of in vitro and field antagonism assessments, endophytic colonisation assessments, and plant performance measurements were used to show that *E. nigrum* B14-1 has the potential to safely antagonise *N. ditissima* in field conditions.

## 2. Materials and Methods

### 2.1. Isolation and Identification of Endophytic Epicoccum Strains

One year old apple shoots, cv. Royal Gala and Queen Cox, were collected from Wiseman orchard, NIAB, East Malling (51.28644° N, 0.46548° E) in the autumn and winter of 2017–2018. Three 5 mm thick transverse sections were excised from each shoot. Bark and wood were dissected with sterile dissection needles. Epiphytic microorganisms were removed with a surface sterilisation technique based on method no. 3 described in Schulz et al. [31]. Surface sterility was confirmed by imprinting on Potato Dextrose Agar (PDA; Oxoid Ltd., Basingstoke, UK). Samples were plated on PDA amended with 40 ppm iprodione (Rovral® WG, UK) and 20 ppm rifamycin (Sigma-Aldrich, Darmstadt, Germany) and incubated at room temperature. Colonies were sub-cultured and maintained on PDA without antibiotics. Candidate *Epicoccum* isolates were selected based on colony morphology (fast-growing, suede-like to downy colonies, initially yellow to orange, red or pink, becoming greenish brown to black with age) and spore morphology (large, globose, and darkly pigmented) [32]. The DNA of candidate strains was isolated from mycelia using the Extract 'n' Amp kit (Sigma). PCR amplification of the ITS region (primers ITS1/ITS4 [33]), the transcription elongation factor (TEF) gene (primers EF1-728F/EF-2R [34]), and the actin (ACT) gene (primers ACT-512F/ACT-783R [35]) was followed by Sanger sequenc-

ing. The nucleotide-BLAST web service (NCBI) was used to retrieve closely related sequences. Geneious Prime 2022.0.1 (https://www.geneious.com, accessed on 27 November 2022) was used to align the sequences and build phylogenetic trees to infer taxonomy, using the Jukes-Kantor distance model with neighbour-joining tree building method using 1000 bootstrap iterations.

### 2.2. In Vitro Antagonism Assays against N. ditissima

The antagonistic capacity of *Epicoccum* strains was assessed in vitro against three *N. ditissima* strains (Hg199, R6/17_3, and R28/15). The reduction of *N. ditissima* growth (Rg) was calculated using the following formula:

$$\text{Rg [\%]} = 1 - ((\text{mean } N. \text{ ditissima growth [mm] }_{test})/(\text{mean } N. \text{ ditissima growth [mm] }_{control})) \times 100. \tag{1}$$

All *Epicoccum* isolates were assessed in co-culture assays. Strain B14-1 was the only *Epicoccum* isolate that sporulated profusely under tested conditions (Supplementary Data), a prerequisite for field trials, and was thus selected for further study using light microscopy observations, and volatile and secreted metabolite assays. All in vitro experiments were conducted twice with 3–5 replicates per experiment.

### 2.2.1. Antagonism in Co-Culture Assay

Mycelial plugs (5 mm diameter) were removed from the edges of 10–14-day-old *N. ditissima* and *Epicoccum* spp. PDA plates. Test plates (9 cm diameter, PDA) consisted of one plug of *N. ditissima* and one plug of test strains separated by 6 cm. Control plates were configured the same way, but without the test strain plug. Five to ten replicates were used per *N. ditissima* strain/test strain combination, and corresponding controls. Plates were incubated at 20 ± 2 °C with 12 h day conditions, and the growth of *N. ditissima* along the shortest path between the pathogen and the test plug was measured at 14 days. Differences in *N. ditissima* growth between test and control plates were determined using a two-sided t-test in Microsoft Excel.

The B14-1—*N. ditissima* interaction on co-culture plates was observed via light microscopy. Briefly, mycelia sections on the PDA medium were excised from co-culture plates, transferred to a microscopy slide, and observed with an inverted microscope (Leica, DM8000). Hyphal morphology of B14-1 and *N. ditissima* at the interaction area (the area closest to the opposing strain) was compared to the control area (growing furthest away from the opposing strain on the same plate) at approximately 24 h before and after the hyphae came into direct contact. Three independent plates were observed per *N. ditissima* strain and representative images were taken.

### 2.2.2. Antagonism of Soluble Metabolites

Eighteen three-week-old B14-1 PDA plates were homogenised in a blender and mixed with chromatography grade ethyl acetate (Sigma) in a 1:2 (mass:volume) ratio and incubated for 12 h in 1 L Duran bottles with constant agitation at 80 rpm. The solvent was filtered through 4 layers of muslin, followed by centrifugation at 4500× *g* for 15 min. The supernatant (200 mL) was evaporated at 45 °C under vacuum and the residuals were resuspended in pure dimethyl sulfoxide (DMSO) (Sigma). The antagonistic effect of ethyl acetate soluble B14-1 metabolites on *N. ditissima* mycelial growth was tested on PDA plates with pure DMSO as a control. One 5 mm plug of *N. ditissima* was positioned in the centre of the PDA plate and incubated for seven days for the colony to reach approximately 2 cm in diameter. On each plate, 25 μL of B14-1 extract and control DMSO were added on opposite sides of the colony at exactly 1 cm from the edge of the growing *N. ditissima* mycelia. Five replicate plates were prepared per *N. ditissima* strain. Growth of *N. ditissima* mycelia towards control and B14-1 extracts was measured on each plate five days after set up. The effect of B14-1 extract on *N. ditissima* growth was determined using a mixed linear model in the 'lme4' R package [36] with "Plate" as a random effect and "Treatment" as a fixed effect.

### 2.2.3. Antagonism of Volatile Metabolites

One 5 mm plug of *N. ditissima* or B14-1 was cut from the edge of a growing mycelial culture, placed onto a fresh PDA plate, and incubated for 2–3 days (20 ± 2 °C). The lids of two plates were removed and the plates were sealed together with Parafilm® (Sigma-Aldrich) to produce a double-agar plate. Test double-agar plates consisted of one *N. ditissima* and one B14-1 plate, and control plates consisted of *N. ditissima* or B14-1 strains on both plates. Six replicates were used for each *N. ditissima*/B14-1 combination and controls. Double-plates were incubated at 22 ± 2 °C with 12 h day conditions. The plates were turned 180° around their horizontal axis every 2–3 days. The diameters of *N. ditissima* and B14-1 colonies were measured at the time of sealing and 16 days after. Statistically significant differences in growth (diameter 16 dpi—diameter 0 dpi) between treatments and controls were determined using a t-test as above.

### 2.3. Field Antagonism Assays

### 2.3.1. Inoculum Preparation

B14-1 spores were produced on peat:lentil:vermiculite:water media in the mass ratio of 1:1:0.5:1 [37]. The media mix (500 g) was autoclaved (30 min at 121 °C) twice in polypropylene bags with 0.5 micron air filter patches (Unicorn, Type 14a). Sterile water was added to the mixture in a 1:1 mass ratio after autoclaving. Bags were inoculated with mycelial plugs from 14-day-old *E. nigrum* PDA plates at a rate of one whole plate per bag and incubated at 22 ± 2 °C in 12 h day conditions. Media was uniformly colonised 2–3 weeks after inoculation and the spores were harvested five weeks after inoculation. Sporulating media was mixed with distilled water and shaken vigorously for 5 min, homogenate was filtered through two layers of sterile muslin cloth and centrifuged at 4500× *g* for 15 min. The spore pellet was then resuspended in sterile distilled water.

The three strains of *N. ditissima* were cultured on sugar nutrient agar with yeast medium (SNAY) [38] at room temperature for six weeks. *N. ditissima* conidia were harvested by crushing sporodochia in sterile distilled water. Conidia of the three *N. ditissima* stains were pooled and the total concentration of macroconidia (two or more cells) was determined with a haemocytometer.

*N. ditissima* and B14-1 spore germination were estimated by incubation in water for 48 h. Spores with visible germination tubes longer than the width of the spore were considered viable. Inoculum was used within 6 h of preparation.

### 2.3.2. Leaf Scar Protection

Leaf scar protection was conducted on mature apple trees cv. Royal Gala (Wiseman orchard, NIAB East Malling, 51.28644° N, 0.46548° E). The leaf scar experiment was designed to a) assess the biocontrol potential of B14-1, and b) to measure its effect on the host, via bud survival upon B14-1 inoculation. A fully randomised field trial was conducted over two separate days in November 2019 (at 30% leaf fall). Each day 24 trees were inoculated across six rows (four trees per row), for a total of 48 trees over 12 rows. Fresh *N. ditissima* and B14-1 inoculum were prepared each day. Four one-year-old shoots were randomly selected on each tree. Four random leaf scars (at least three scars apart) were freshly exposed on each shoot and randomly assigned to one of the following treatments: (1) water control, (2) *N. ditissima* only ($7 \times 10^4$ conidia per ml in water), (3) B14-1 only ($8 \times 10^5$ conidia per ml in water) and (4) B14-1 + *N. ditissima* (B14-1 conidia followed by *N. ditissima* conidia within 20 min). Water and conidial suspensions were applied at 5 µL per leaf scar.

The incidence of (a) *N. ditissima* canker lesions (b) dead buds (leaf scars with no canker lesion and not producing foliage from the adjacent bud), and (c) healthy leaf scars (with the adjacent bud producing healthy foliage) was recorded in June 2020, approximately eight months after inoculation. The size of canker lesions was also measured as the longest lesion length along the branch.

### 2.3.3. Pruning Wound Protection

B14-1-mediated protection of pruning wounds was tested across the same two days, on the same 48 Royal Gala trees as above using a separate set of eight random shoots per tree. Pruning wounds (one per shoot) were created by cutting each shoot 10 cm below the apex with clean secateurs. Four shoots were randomly selected and treated with 30 μL of B14-1 conidia at $1 \times 10^5$ per ml in water; the remaining control wounds were treated with 30 μL of water. The next day, all pruning wounds were inoculated with 30 μL of *N. ditissima* at $10^4$ conidia per ml. *N. ditissima* canker lesion incidence was recorded on pruning wounds in June 2020, approximately eight months after inoculation.

### 2.3.4. Field Data Analysis

The incidence of each leaf scar outcome (canker, dead, or healthy) and canker lesion outcome on pruning wounds were summarised per tree (split-plot), with each of the 12 rows forming a block. The odds ratios of each outcome were analysed separately using a generalised linear mixed model (binomial distribution, logit link function) in the 'glmmTMB' R package [39]. Tree nested within a block nested within a day ("Day/Block/Tree") was used as a random effect to account for the split-plot design over multiple blocks and inoculation days, and "Inoculation treatment" was used as a fixed effect.

### 2.4. Endophytic Colonisation of B14-1

Experimental design and sampling of apple leaf scars inoculated with B14-1 in autumn.

Short-term colonisation of shoots was assessed in an independent experiment at leaf fall in October 2020. Apple tree (cv. Royal Gala) leaf scars were inoculated with B14-1 spores as above. Heat-inactivated B14-1 conidia were used as a control to ensure that DNA arising from inoculum was accounted for in the qPCR assay. The experiment consisted of five blocks, each with paired plots (B14-1 and control), and three trees per plot. At 10 and 20 days post inoculation (dpi) the concentration of B14-1 was assessed in the leaf scar tissue and also in the tissue above and below the leaf scar to assess its systemic spread. To ensure the samples were as representative as possible, tissues from two shoots per tree and from three trees within a plot were pooled into a single sample (six shoots total). Shoots were removed with secateurs and cleaned using three cycles of: brushing with a clean brush, rinsing in tap water, and wiping with 70% ethanol. Two leaf scars per shoot were dissected into (a) leaf scar tissue, as in Olivieri et al., 2021 [22]; (b) bark and sapwood from 3–5 mm; and (c) 8–10 mm above and below the leaf scar. Sections were sampled by making 3 mm deep cuts at 3, 5, 8, and 10 mm above and below the leaf scar using a sterile scalpel and collecting the appropriate semi-circular sections (12 sections from six shoots over three trees per sample).

Long-term colonisation of B14-1 in apple tissues was investigated using canker lesion-free leaf scars from the 2019 leaf scar protection experiment (see above), sampled in October 2020, one year after inoculation. Each sample consisted of 12 tissue sections belonging to the same inoculation type (B14-1 or water control) and tissue segment (leaf scar, 3–5 mm or 8–10 mm above and below a leaf scar), pooled together from four trees in the same row (three sections per tree). Four rows were sampled for four independent replicates of each inoculation/tissue type combination. DNA was extracted and *E. nigrum* DNA abundance in the samples was measured with qPCR as described above.

### 2.4.1. Quantification of *E. nigrum* DNA in Plant Tissues

Apple tissue samples were frozen at $-20\,^\circ$C, freeze-dried for 48 h, and homogenised with a Geno/Grinder2010 (SPEX SamplePrep, Stanmore, UK) at 1500 rpm for 90 s using steel ball bearings (5 mm diameter). Homogenised samples were resuspended in sterile phosphate buffer saline (pH 7.5) in a 1:5 dry weight to volume ratio and stored at $-20\,^\circ$C. DNA was extracted from 120 μL of resuspended samples using the Qiagen Plant Mini DNA Extraction Kit (Qiagen, Germany) following standard protocol [22]. The abundance of *E. nigrum* in the samples was measured using quantitative PCR (qPCR) with primers

and probes from the '*Epicoccum nigrum* 18S Ribosomal RNA Gene Detection Kit' (Genesig, Primer design Ltd.). According to the manufacturer's instructions, the kit should detect the majority of known strains of *E. nigrum*. Comparison of qPCR results obtained from DNA extracts of pure B14-1 with the absolute standard curve supplied with the kit showed that the qPCR was able to reproducibly detect B14-1 down to 10 copies per reaction. SsoAdvanced Universal Probes Supermix (Bio-Rad, Cambridge, MA, USA) and CFX384 Touch Real-Time PCR Detection System (Bio-Rad) were used to run 10 μL reactions with 2 μL of sample DNA and primer/probe concentrations according to the manufacturer's instructions. Each sample was tested both undiluted and 10× diluted, and each dilution was tested in triplicate. *E. nigrum* 18S RNA gene copy number per 1 μL of extracted DNA was calculated by comparing sample Cq value with the absolute standard curve supplied with the qPCR kit and adjusting for dilution factor.

### 2.4.2. Colonisation Data Analysis

Data was log10 transformed to fit a normal distribution. Short-term colonisation (10 and 20 dpi) was analysed using a linear mixed model with a normal distribution in the 'lme4' R package [36]). "Plot" was used as a random intercept to account for the fact that the same plots were re-sampled (a) for different tissue sections and (b) at different dpi. "Block", "Inoculation treatment", "Tissue section" and "Day post inoculation" were used as fixed effects. Survival of B14-1 one year after inoculation was analysed using ANOVA with "Inoculation treatment", "Tissue section" and "Row" as fixed effects.

### *2.5. Investigating Potential Adverse Effects of B14-1*
### 2.5.1. Effects of B14-1 on Apple Tree Growth

We tested whether the summer application of B14-1 affects the growth of apple M9 rootstocks in stool beds. Stool beds were located in the East Egham plot, NIAB East Malling (51.28728° N, 0.45683° E). Rootstock stool beds were used due to the ease of spore application to roots. Two independent experiments were conducted in the summers of 2018 and 2019, each on a separate rootstock stool bed. Each experiment was conducted on four randomised blocks with one of the following treatments on each of four plots per block: (1) water control, (2) B14-1 foliar spray, (3) B14-1 root drench, and (4) B14-1 root drench + foliar spray. Each plot consisted of 10 rootstock plants with 3–5 shoots per plant. B14-1 inoculum ($2.4 \times 10^6$ spores per plant) was applied in the late afternoon as a water solution at a rate of 100 mL per plot. In December 2018 and 2019 the rootstocks from the corresponding trial were harvested, graded by stem diameter (at 15 cm above the soil line) into three categories: small (<9 mm), medium (9–11 mm), and large (>11 mm). The overall effect of B14-1 augmentation on the frequency of each diameter class (small, medium, and large) was analysed using an ordinal regression model in the 'ordinal' R package with "Year", block within a year ("Year:Block") and "Treatment" as fixed effects.

### 2.5.2. Effects of B14-1 on Apple Fruit

Mature apple fruits cv. Gala, Braeburn and Jazz (Kent, UK origin) were bought in the local supermarket, washed in tap water, wiped with 70% ethanol, and dried in a laminar flow hood. Fruits were pierced five times 1 mm wide and 2 mm deep with a sterile hypodermic needle, randomly assigned to B14-1 and control groups (five fruit per group per cultivar), and sprayed to just before run-off with a B14-1 spore suspension ($10^5$ conidia per ml) or water (control) using a hand atomiser. Fruits were incubated for four weeks in high humidity trays at 22 °C with 12 h light. Wounded and intact parts of B14-1 and control fruits were inspected for atypical rot. The experiment was repeated twice, in February and March 2022.

### *2.6. General Statistical and Visualisation Tools*

R packages required for analysis were run in RStudio Version 1.4.1717 (R version 4.1.1). Type I analysis of variance (ANOVA) with Satterthwaite's method and type II ANOVA

with Wald Chi-square statistics were used for linear mixed and generalised (binomial) linear mixed models, respectively. Model diagnostics were performed in the "DHARMa" package [40]. Estimation of treatment means, standard errors, and pairwise post hoc-testing was performed using the "emmeans" package [41] with default settings. The data were visualised with the "ggplot2" [42] and "ggpubr" [43] packages.

## 3. Results

### 3.1. Isolation and Identification of Epicoccum Strains

Out of 287 isolated apple fungal endophytes, three (Table 1) showed colony morphology on PDA plates and conidia morphology analogous to *E. nigrum*., i.e., fast-growing, dense areal mycelium ranging from yellow to orange-brown in colour with multicellular, globose and darkly pigmented conidia that were 15–25 μm in diameter with a funnel-shaped base and verrucose external surface (Figure 1). The ITS sequences of the three strains showed over 99.9% similarity with published sequences of *E. nigrum*, *E. layuense* and *E.tritici*. Of the three isolated strains, there was one nucleotide difference between ITS sequences of B14-1 and C29 and between C15 and C29, and two between B14-1 and C15. Additional sequencing and phylogeny analysis of the TEF and ACT regions of isolate B14-1 showed clear and consistent clustering with *E. nigrum* representative strains (Supplemental Figures S1 and S2).

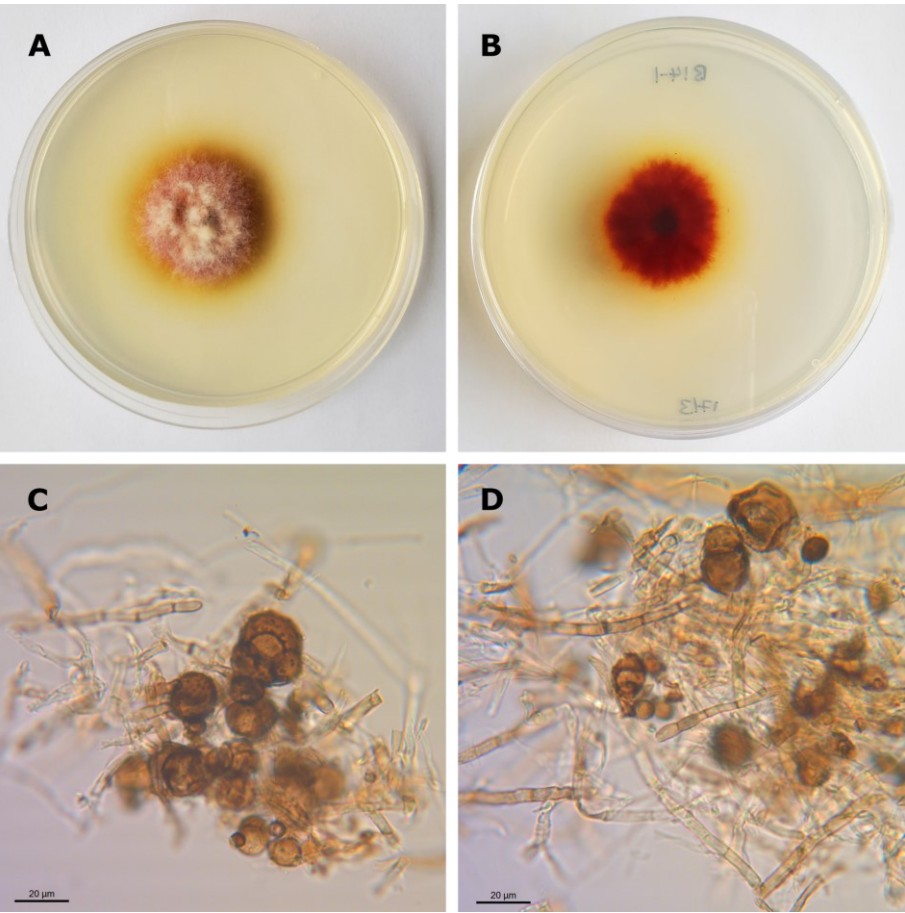

**Figure 1.** *E. nigrum* B141 colony, mycelium, and spore morphology. Front (**A**) and back (**B**) of of 5-day old B14-1 culture on PDA Representative microscopy images of B14-1 mycelium fragments and conidia at different maturity stages (**C**,**D**).

**Table 1.** Taxonomic and in vitro antagonism data of endophytic *Epicoccum* spp. strains isolated in this study. Mean growth reduction (%) v s. the control $\pm$ 1 SE is shown across three different *N. ditissima* isolates and two experiments, with 5–10 replicates per experiment. Isolates C15 and C29 were not tested (n.t.) in volatile and metabolite assays. Statistically significant ($p$ ($t$-test) < 0.01) reduction of growth compared to the control is denoted with *.

| *Epicoccum* **spp. Isolate** | | **B14-1** | **C15** | **C29** |
|---|---|---|---|---|
| Isolated from Apple cv.: | | Royal Gala | Queen Cox | Royal Gala |
| NCBI accession: | ITS | OM650677 | OM650678 | OM650679 |
| | TEF | OP753363 | n.t. | n.t. |
| | ACT | OP753362 | n.t. | n.t. |
| Taxonomic classification | | *Epicoccum nigrum* | *Epicoccum* spp. | *Epicoccum* spp. |
| Reduction of *N. ditissima* growth (%) in vitro: | Co-culture assay | 52.0 $\pm$ 1.4 * | 45.7 $\pm$ 1.5 * | 55.8 $\pm$ 2.6 * |
| | Soluble metabolite assay | 20.0 $\pm$ 2.2 * | n.t. | n.t. |
| | Volatile metabolite assay | 3.1 $\pm$ 1.4 | n.t. | n.t. |

### 3.2. In Vitro Antagonism

All three strains reduced *N. ditissima* growth in vitro in dual co-culture assays on PDA plates by ca. 50% (Table 1, Figure 2A). Furthermore, the tips of *N. ditissima* hyphae were observed growing away from B14-1 at an approximately 90° angle before the two organisms came into direct contact on co-culture plates (Figure 2E). B14-1 hyphae did not change direction, but upon contact with *N. ditissima*, the tips changed morphology from uniformly thin with few branching points (Figure 2C) to thick, bulbous and heavily branching hyphae (Figure 2F). *N. ditissima* hyphae did not visibly change their morphology upon contact with B14-1. Soluble metabolites produced by B14-1 reduced *N. ditissima* growth on PDA plates by ca. 20% (Figure 2B, Table 1). Volatile metabolites assayed in the double-agar plate test had no effect on *N. ditissima* growth on PDA. Interestingly, B14-1 grew slightly, but significantly faster when sharing the atmosphere with *N. ditissima*; 9% larger colony diameter at 9 dpi when growing in a double-agar plate with *N. ditissima* in comparison to its control, i.e., plates with B14-1 on both sides.

### 3.3. Antagonism in Field Conditions

The incidence of *N. ditissima* lesions on leaf scars was significantly affected by inoculation treatment (Chisq = 30.11, df = 3, $p = 1.3 \times 10^{-6}$). Canker incidence was reduced by 46.2% when leaf scars were co-inoculated with B14-1 (32.6% incidence) in comparison to *N. ditissima* alone (60.6% incidence) (Tukey post hoc $p = 4.18 \times 10^{-6}$) (Figure 3A). The difference in canker lesion incidence resulting from natural infections of water (2.8%) and B14-1 only treatments (8.6%) was not significant. The size of canker lesions on leaf scars was not affected by inoculation treatment. Co-inoculation with B14-1 and *N. ditissima* resulted in a significantly higher incidence of healthy leaf scars (52.6%) in comparison to *N. ditissima* alone (28%) (Tukey post hoc $p = 3.5 \times 10^{-5}$ (Figure 3C). B14-1 inoculation did not increase the incidence of dead buds nor decreased the incidence of healthy leaf buds in comparison to the water control (Figure 3B,C). In fact, B14-1 inoculated leaf scars had a higher incidence of healthy leaf scars and a lower incidence of dead buds compared to the water control, but the difference was not significant. Representative phenotypes of leaf scars in each treatment are shown in Supplemental Figure S3. Canker incidence was significantly reduced (by 5.5%) on B14-1 protected pruning wounds (90.7%) compared to unprotected wounds (96%) (Chisq = 4.88, df = 1, $p = 0.027$) (Figure 3D). Block (row in the orchard) and inoculation day did not affect canker symptom development in leaf scar or pruning wound protection or experiments.

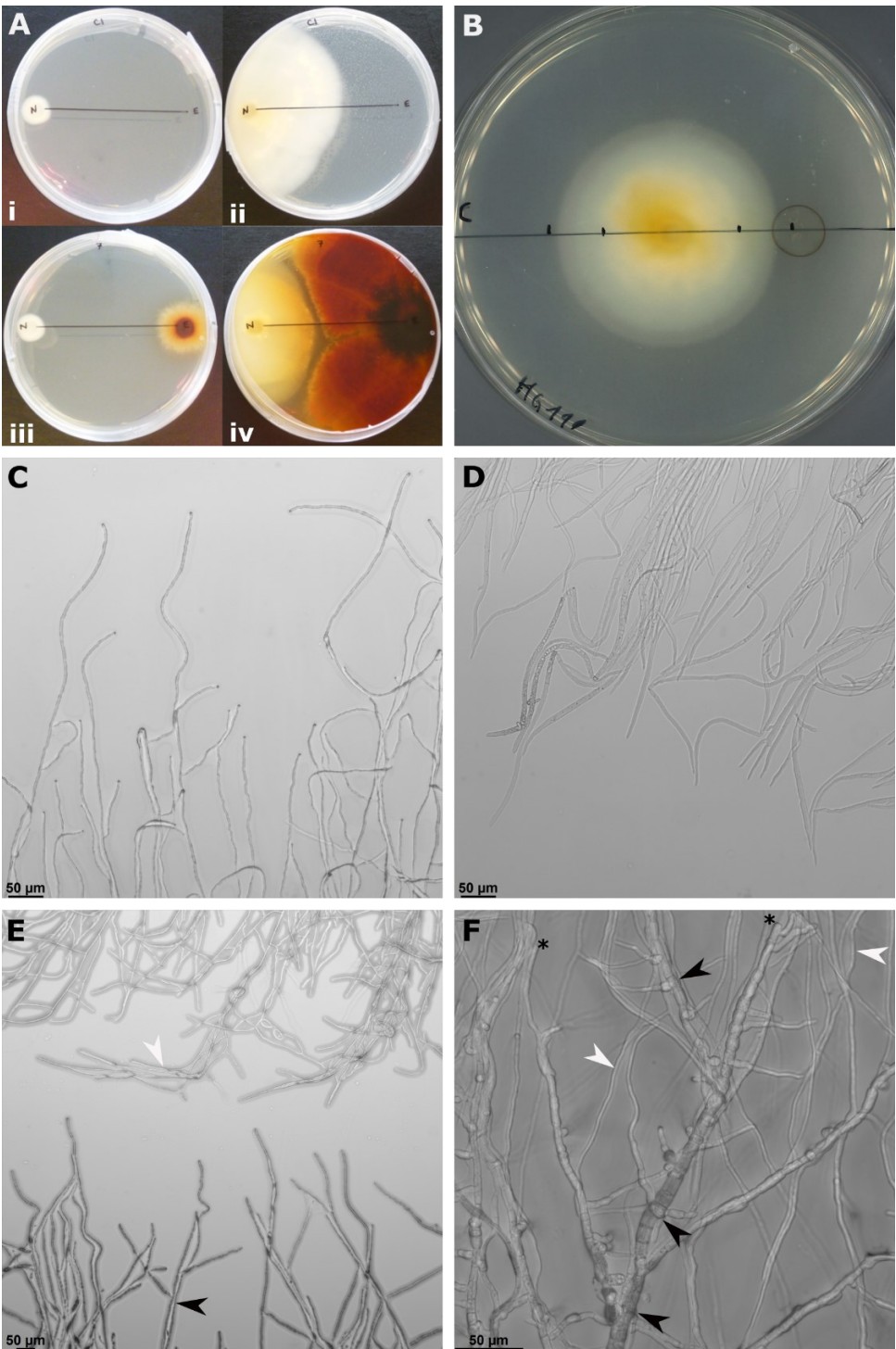

**Figure 2.** In vitro biocontrol assays. Co-culture assay (**A**) showing *N. ditissima* control plates (**i,ii**) and B14-1 co-culture test plates (**iii,iv**) at 3 (**i,iii**) and 14 days (**ii,iv**) after assay set up. Soluble metabolite assay (**B**) with DMSO control on the left-hand side (labelled as "C") and B14-1 extract on the right-hand side of the panel. Microscopy observation of co-culture plates (**C–F**). Morphology of B14-1 (**C**) and *N. ditissima* (**D**) hyphae when grown in isolation. On co-culture plates *N. ditissima* hyphae (**E**, white arrow) were observed growing at an almost 90° angle to B14-1 hyphae (**E**, black arrow) immediately before contact with B14-1. Upon contact, the tips of B14-1 hyphae became bulbous, thickened, and increased branching (**F**, black arrows). *N. ditissima* hyphae (**F**, white arrows) did not change morphology upon contact. The tips of B14-1 hyphae in panel **F** are denoted with *.

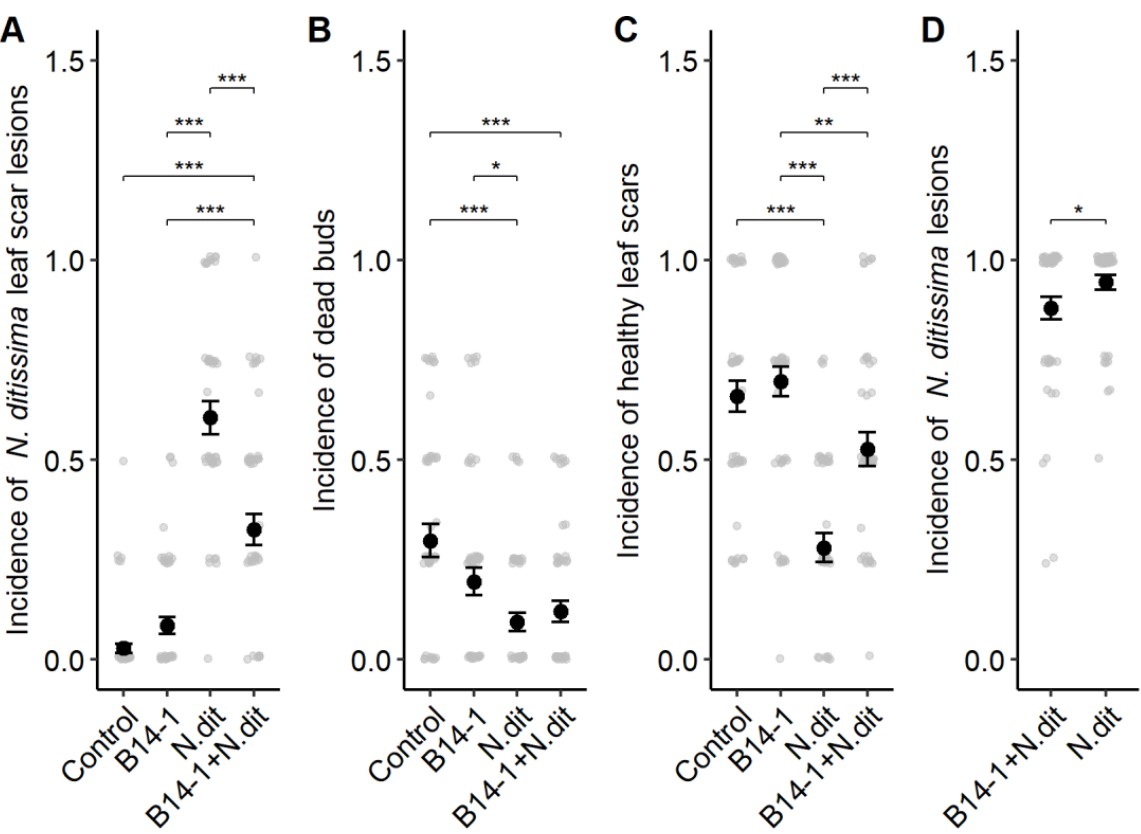

**Figure 3.** Antagonistic efficacy of B14-1 against *N. ditissima* in the field trial. Outcomes of the leaf scar protection (**A–C**) and pruning wound protection experiments (**D**) are shown (n = 48). Incidences of *N. ditissima* canker lesions (**A,D**), dead buds without a clear canker lesion (**B**) and of healthy leaf scars where adjacent buds produced healthy foliage (**C**) are presented. Treatments are shown along the horizontal axis, namely, control leaf scars (water) are shown alongside inoculated with B14-1 (B14-1), *N. ditissima* (N.dit) alone, or a combination of both (B14-1 + N.dit). Estimated marginal means ± 1 SE and raw data points (in grey, n = 48) are displayed. Statistical significance between the treatments is presented as: "*" (0.05 > *p* > 0.01), "**" (0.01 > *p* > 0.001) and "***" (*p* < 0.001).

### 3.4. Colonisation of Apple Tissues

Analysis of short-term colonisation (10, 20 dpi) of apple shoots upon B14-1 leaf scar inoculation showed that sampling block (F = 3.61, df = 4, *p* = 0.01), inoculation treatment (F = 8.79, df = 1, *p* = 0.005) and sampling tissue section (F = 27.69, df = 2, *p* < 0.0001) significantly affected *E. nigrum* abundance in apple tissues. *E. nigrum* abundance in apple shoot tissues was slightly higher at 20 dpi than at 10 dpi, but the difference was not significant (F = 3.26, df = 1, *p* = 0.08). Comparing treatment vs. control samples (across all sampling tissues and times) indicated that a single inoculation with live B14-1 spores increased short-term *E. nigrum* abundance by approximately 1.53 times compared to the dead spore control. *E. nigrum* abundance was the highest in leaf scar tissue, while its concentration dropped by approximately 10-fold in samples 3–5 mm and 8–10 mm away from the inoculated leaf scars (Figure 4). B14-1 inoculated tissues had a higher mean *E. nigrum* abundance compared to the control in every tissue/timepoint combination, but the differences were not pairwise significant (Tukey post-hoc pairwise comparison). This is most likely due to a high number of individual comparisons, large variation within groups, and relatively low subgroup sample size (n = 5).

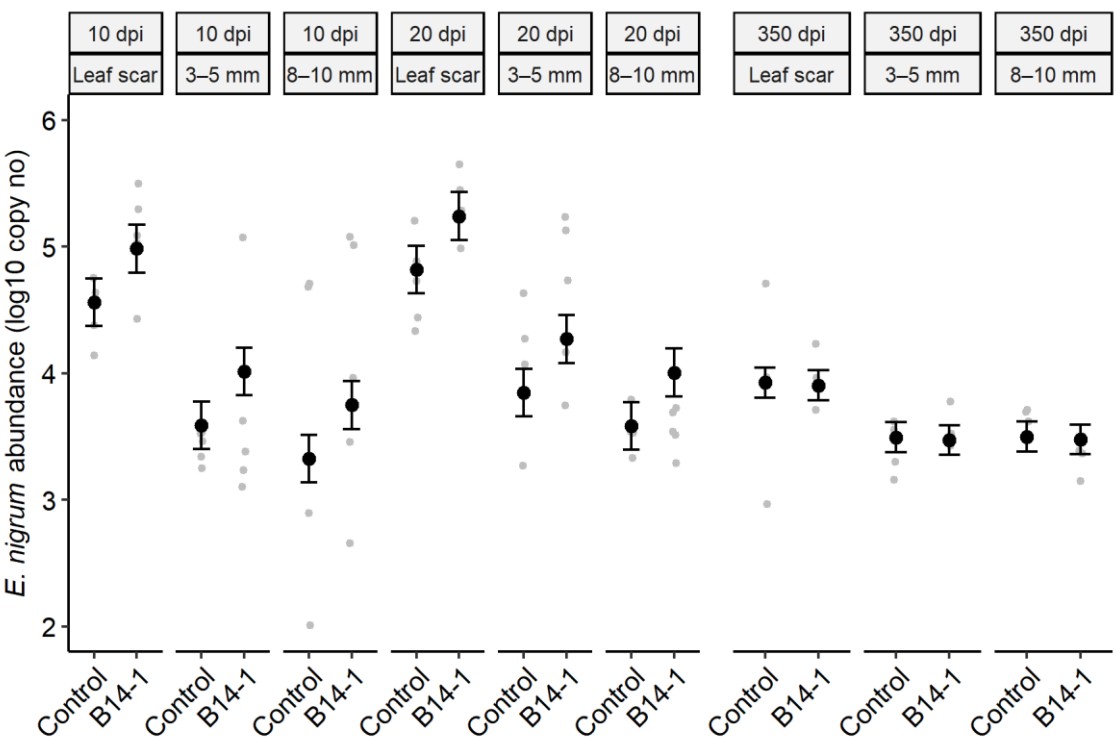

**Figure 4.** The abundance of *E. nigrum* DNA in shoots of cv. Royal Gala upon leaf scar inoculation. Estimated marginal means of abundance (total copy number per 1 μL extracted DNA) ± 1 SE in leaf scar tissue ("Leaf scar"), in bark wood 3–5 mm above and below ("3–5 mm") and 8–10 mm above and below the leaf scar ("8–10 mm"), are shown at 10, 20 and 350 dpi. The treatments, presented on the horizontal axis, were: live B14-1 spores (B14-1), inactivated B14-1 spores (Control at 10, 20 dpi), or water (Control at 350 dpi). Raw data points for each time/tissue/treatment combination are shown in grey (10, 20 dpi n = 5; 350 dpi, n = 4). Each data point represents a pool of 12 leaf scars or bark and wood sections from three (10, 20 dpi) or four trees (350 dpi). The groups at 350 dpi are from a different experiment and are not directly comparable with 10 and 20 dpi.

*E. nigrum* abundance one year after B14-1 inoculation (350 dpi) was not affected by block or inoculation treatment but it was still affected by tissue section (F = 5.87, df = 2, $p$ = 0.01). Overall *E. nigrum* abundance was similar in inoculated and control samples (Figure 4).

### 3.5. Potential Adverse Effects of B14-1 on Apple

No necrotic or chlorotic symptoms were observed on the foliage of B14-1 treated M9 rootstock in either of the experimental years. The foliage of M9 rootstocks treated with B14-1 was phenotypically indistinguishable from the control (see representative images of M9 foliage in Supplemental Figure S3). The growth of rootstocks measured as rootstock stem diameter was not affected by B14-1 augmentation (Supplemental Figure S4), i.e., was comparable to the control.

Careful inspection and comparison of wounded and intact fruit surfaces of control and B14-1 inoculated Gala, Braeburn, and Jazz fruit did not indicate any symptoms due to B14-1 at four weeks of room temperature incubation (Figure 5).

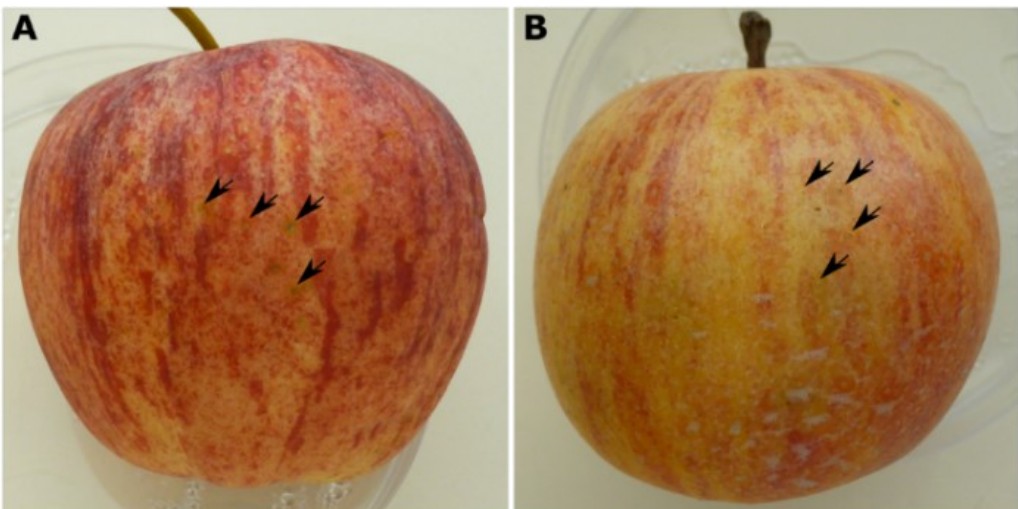

**Figure 5.** Representative images from water control (**A**) and B14-1 (**B**) treated cv. Gala fruit after four weeks of incubation at 22 °C. Needle wounds are highlighted with arrows.

## 4. Discussion

In this study, ITS metabarcoding guided the screening of apple endophytes and enabled rapid identification of *E. nigrum* strain B14-1 as a potential biocontrol agent of *N. ditissima*. B14-1 showed antagonism against *N. ditissima* in vitro and also in commercial field conditions. B14-1 was able to temporarily increase *E. nigrum* abundance in apple shoots following autumn leaf scar inoculation in field conditions, but the increase did not persist from one autumn to the next. Importantly, B14-1 did not cause any growth penalty or disease symptoms on leaves, shoots, buds, or fruits. This is, to our knowledge, the first proof of principle study of the efficacy, colonisation, and possible undesired effects of an endophytic fungal biocontrol agent against apple canker in commercially relevant conditions.

Rather than screening hundreds of candidates [14], this study was the first to use information from ITS amplicon sequencing [6] of healthy tree endophytes to focus biocontrol discovery on candidates with high antagonism and colonisation potential, and low risk to the host plant. This approach, however, is only useful if the ITS amplicon sequencing identifies known candidates, and can only be very effective if candidates are morphologically distinct as are the *Epicoccum* spp. strains found in this study.

Isolates *of Epicoccum* spp. were previously found to act as biological control agents against the Esca disease complex of grapevine [44], *Hymenoscyphus fraxineus* [45], *Candidatus phytoplasma mali* [46], *Cytospora cincta* [46] and *Monilinia laxa* [27,47]. Strains of *Epicoccum* spp. with in vitro biocontrol potential against *N. ditissima* have been isolated from apple trees in New Zealand [23]. This study has shown that endophytic *E. nigrum* B14-1 antagonised *N. ditissima* in vitro through the secretion of soluble, but not volatile, antimicrobial compounds, and fast and responsive growth likely leading to effective competition for nutrients. Secretion of antagonistic soluble metabolites in vitro is in line with previous reports on *E. nigrum* strain P16 [26] and was corroborated in this study by microscopic observation of *N. ditissima* hyphae avoiding B14-1 contact on PDA. Identification of novel antimicrobial metabolites secreted by B14-1, though an interesting and important area for research, was out of the scope of this study. The increased growth rate of B14-1 when exposed to *N. ditissima* volatile compounds and changes to hyphal morphology (thickening and branching) indicated that B14-1 perceived *N. ditissima* and responded by more aggressive growth.

Research and development of biological control agents is not a straightforward process. Many biocontrol studies carried out under controlled conditions circumvent complications associated with environmental influences and large-scale production of inoculum [48]. *Bacil-*

*lus subtills* has been previously reported to control *N. ditissima* infections on leaf scars [49], but none of the commercial biocontrol products tested so far have been effective against *N. ditissima* in the field [13,50]. This study showed significant protection of leaf scars and pruning wounds against *N. ditissima* in commercially relevant leaf fall conditions for the first time. Our data indicated that B14-1 protected apple tissues from *N. ditissima* entry, shown by a reduction in incidence, but not from its spread upon successful infection, as there was no effect on canker size. To corroborate our findings, the experiments will need to be repeated with other commercially relevant apple cultivars and, if possible, with commercially relevant spray application(s) of B14-1 in conjunction with natural inoculum pressure. Interestingly, endophytic *Epicoccum* strains were not found in high abundance in the most recent fungal community analysis of leaf scars in three UK orchards [22,51], highlighting the scope for increasing *Epicoccum* abundance to improve apple canker resilience.

Additionally, we have obtained encouraging preliminary in vitro findings showing that B14-1 mycelia and metabolites could in vitro antagonise *Phytophthora cactorum*, an oomycete pathogen causing fruit and collar rot on apples (Supplemental Figure S5). These findings indicate that B14-1 could have biocontrol potential against a range of apple pathogens, increasing its commercial impact.

Colonisation strategy and establishment in the host is an important considerations for endophytic biocontrol which has been largely overlooked [48]. In this study, we used qPCR to track the total *E. nigrum* population in apple shoots. Using appropriate controls and a replicated randomised experimental design, we assumed that the measured increase in the total *E. nigrum* community size was mostly due to B14-1. The study of apple shoot colonisation upon leaf scar inoculation revealed that, unlike *E. layuense* E24 in grapevine [44], *E. nigrum* B14-1 did not extensively colonise apple tissues above or below inoculated leaf scars within the first 20 days and did not persist in apple shoots across the two tested seasons. This indicated that several doses of B14-1 applied every year will likely be required for effective biocontrol of *N. ditissima*. To our knowledge, this is the first report where the colonisation success of an endophytic biocontrol fungus was assessed on a woody perennial host in commercially relevant conditions. The B14-1 colonisation of apple tissues may have been affected by the apple cultivar used and experimental conditions such as resident microbiome and climate conditions during and after inoculation. The baseline levels of *E. nigrum* in our experimental trees were fairly high, with well over 1000 ITS copies per μL of DNA extract. *E. nigrum* strains have the potential to antagonise each other [52], which means that naturally present *E. nigrum* strains may have reduced the success of B14-1 establishment. To verify our findings the colonisation experiments will need to be repeated using other commercially relevant cultivars and orchards with different *E. nigrum* baseline levels. It is likely that host genotype plays a role in B14-1 colonisation since ca. 10× higher abundance of *Epicoccum* OUTs was found in less susceptible apple scion varieties (e.g., Golden delicious) compared to more susceptible (e.g., Gala) [6]. B14-1 colonisation may therefore be better and longer lasting in less canker-susceptible scions. It is also possible that B14-1 needs to establish itself on leaf surfaces and gradually colonise leaf tissue to reach leaf scars as reported for *E. nigrum* P16 on sugarcane [26]. It may be that spring or summer applications of B14-1 to foliage could yield better leaf scar colonisation than the autumn application tested here. More research, however, is required to optimise the B14-1 formulation and determine its host genotype and tissue preference, optimal application timing and conditions, and seasonal population dynamics on apples and the wider orchard environment. Using fluorescently labelled B14-1 could be used to verify the molecular quantification data and reveal further details on the colonisation of B14-1 in different organs and tissues.

In contrast to its reported biocontrol potential, *E. nigrum* isolated from cantaloupe has been shown to cause post-harvest decay on apple, pear, cucumber and tomato [53]. In this study, *E. nigrum* B14-1 did not cause symptoms on M9 foliage or reduce M9 growth when applied as a foliar spray or root drench, and did not alter the health outcomes of leaf buds (cv. Gala) inoculated with B14-1 spores. Moreover, it did not cause symptoms on wounded

or intact mature apple fruit (cv. Gala, Braeburn, Jazz). We thus consider *E. nigrum* B14-1 not detrimental to apples. Our findings are in line with the asymptomatic colonisation of *E nigrum* in sugarcane [26] and *E. layuense* in grapevine [44]. Further investigations are required to ascertain the impact of high B14-1 abundance on the fitness of wild plant, animal, and microbial communities in and around the orchards. The effect of increased B14-1 abundance on the severity of other pathogens [54] and on orchard productivity will also need to be investigated.

## 5. Conclusions

In conclusion, this is the first study reporting efficacy, colonisation, and potential detrimental effects of a fungal biocontrol (*E. nigrum* B14-1) against the devastating apple pathogen *N. ditissima.* This study leveraged metagenomics data to rapidly isolate candidate *Epicoccum* spp. endophytes that are (a) likely to have biocontrol potential against *N. ditissima*, (b) likely to be effective apple colonisers, and (c) unlikely to be pathogenic to the host. Indeed, *E. nigrum* strain B14-1 was shown to control *N. ditissima* in vitro via fast, responsive growth and secretion of antimicrobial compounds. More importantly, B14-1 protected small and large entry points from *N. ditissima* symptomatic infection in a commercial orchard at leaf fall and was able to colonise apple shoots when applied directly to leaf scars in autumn. B14-1 was not pathogenic on leaves, stems, or fruit and was therefore considered suitable for use in commercial orchards. Application formulation, frequency and timing, and effects on non-apple species, and orchard productivity will need to be elucidated before B14-1 could become a part of the integrated management of European apple canker.

**Supplementary Materials:** The following supporting information can be downloaded at: https://www.mdpi.com/article/10.3390/agriculture13040809/s1, including: Preliminary *E. nigrum* sporulation assessment summary; Figure S1: Neighbour-joining tree based on partial actin sequences; Figure S2: Neighbour-joining tree based on partial transcription elongation factor sequences; Figure S3: Representative plant phenotypes observed on Gala leaf scars and M9 rootstocks augmented with B14-1 and controls.; Figure S4: Rootstock diameter class data across two experiments; and Figure S5: B14-1 antagonistic effects on *Phytophthora cactorum* in vitro.

**Author Contributions:** Conceptualisation, X.X., R.S. and M.P.-R.; methodology development, R.S. and M.P.-R.; experiments conducted by M.P.-R., L.O., T.P., J.K., G.F. and H.M.; validation, T.P., J.K. and G.F.; formal analysis, M.P.-R. and X.X.; writing—original draft preparation, M.P.-R.; writing—review and editing, X.X., M.P.-R., L.O., T.P. and G.F.; visualisation, M.P.-R.; supervision, X.X., R.S. and M.P.-R.; project administration, X.X. and M.P.-R.; funding acquisition, X.X. All authors have read and agreed to the published version of the manuscript.

**Funding:** This work was supported by the UK Biotechnology and Biological Science Research Council (BBSRC) [grant number: BB/P007899/1] and several industry organisations: Agriculture and Horticulture Development Board (AHDB), Adrian Scripps Limited, Avalon Produce Limited, ENZA [T&G global subsidiary], Frank P Matthews Limited, and Worldwide Fruit Limited. Leone Olivieri was further supported by AHDB via PhD studentship grant CP161: Understanding Endophytes to Improve Tree Health.

**Institutional Review Board Statement:** Not applicable.

**Data Availability Statement:** Microbial strains, statistical analysis code and raw data are available for research purposes upon request from the corresponding author.

**Acknowledgments:** We thank Nick Dunn at Frank P Matthews, Nigel Jenner at Avalon Produce Ltd. and Tony Harding at Worldwide Fruit Ltd. for providing nursery and grower insights. We are grateful to Neza Molk for her help with biocontrol trials with B14-1 in field conditions.

**Conflicts of Interest:** The authors declare no conflict of interest. The funders had no role in the design of the study; in the collection, analyses, or interpretation of data; in the writing of the manuscript; or in the decision to publish the results.

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
