# Peer review of "From Endophyte Community Analysis to Field Application: Control of Apple Canker (Neonectria ditissima) with Epicoccum nigrum B14-1"

_agriculture, doi:10.3390/agriculture13040809_

Round 1
Reviewer 1 Report
I have been assigned to review the article "From endophyte community analysis to field application: control of apple canker (Neonectria ditissima) with Epicoccum nigrum B14-1". Carefully studying and reviewing the manuscript, several significant issues have been found as mentioned below.
Phenotypes and morphological data are missing for diseased plants supplemented with and without the endophyte.
Endophytic colonization with the host plant should have been confirmed by microscopy.
Epicoccum strain colony phenotypes are missing.
In vitro antagonistic assay phenotypes are missing.
The majority of data have been submitted as supplementary material, however,
I would suggest to include all supplementary data in the main manuscript and perform restructuring of the whole manuscript.
Author Response
I have been assigned to review the article "From endophyte community analysis to field application: control of apple canker (Neonectria ditissima) with Epicoccum nigrum B14-1". Carefully studying and reviewing the manuscript, several significant issues have been found as mentioned below.
Response:
Dear reviewer than you for your comments, suggestions and for pointing out the issues with the manuscript. We have done our best to accommodate all the comments. Our responses are listed point by point below.
- Phenotypes and morphological data are missing for diseased plants supplemented with and without the endophyte.
Response:
We have not included examples of the plant phenotypes with and without B14-1 endophyte amendment in the original manuscript because there were no significantly different phenotypes observed compared to control. We have now added the examples of phenotypes observed in the leaf scar protection experiment and rootstock augmentation experiment to Supplementary figure 3. We have not taken high quality representative images at the time of the trial because there were no differences observed. The images provided in Supp. figure 3 are the best examples we have. Unfortunately, we have no representative images from pruning wound protection experiment.
We would also like to point out that in the original manuscript was have provided data on phenotypes in B14-1 treated and control plants in the form of:
- Quantitative phenotypic data from leaf scar inoculations where incidence of healthy and dead leaf scars was compared between water and B14-1 inoculation (was fig 1, now fig 3 in updated manuscript).
- Quantitative results on M9 rootstock growth with and without B14-1 amendment (lines 421-427 and supp figure 4) and descriptive results in the text (line 422-427) which have now been extended.
- We have provided representative images of apple fruit with and without B14-1 application (was Supp fig 5, now figure 5 in updated manuscript)
- Endophytic colonization with the host plant should have been confirmed by microscopy.
Response:
We agree with the reviewer that microscopy confirmation of colonisation would be ideal, but to do so in apple shoots it would necessitate the generation of fluorescently tagged E. nigrum B14-1 to distinguish it from the resident fungal endophytes. The labelling could have an effect on biocontrol properties and the labelled strain would need to be validate again for its antagonistic potential. Furthermore, fluorescently labelled B14-1 could not be released in the environment and thus its colonisation could not be tested in orchard conditions. We recognise that colonisation experiments with fluorescently labelled B14-1 would be interesting and valuable. They could be done on detached shoots in class 2 containment / controlled environment which was out of scope of this study. We have now commented on this in discussion, see lines 515-517.
- Epicoccum strain colony phenotypes are missing.
Response:
Added to the manuscript as requested.
Images of B14-4 colony morphology on PDA have been merged with microscopy images of spores and mycelium (previously supp fig 1) and are now a part of Figure 1 in the main text.
- In vitro antagonistic assay phenotypes are missing.
Response:
We have previously shown examples of phenotypes observed in antagonistic assays in the supplementary figure which has now been moved to the main text (figure 2). Unfortunately, we have no high quality representative images of phenotypes in volatile assay due to the fact that no differences in N. ditissima growth was observed in comparison to the control. Antagonistic phenotypes have previously been are described in the results section “3.2 In vitro antagonism”. We have now elaborated further on the phenotypes in lines 338-343.
- The majority of data have been submitted as supplementary material, however, I would suggest to include all supplementary data in the main manuscript and perform restructuring of the whole manuscript.
Response:
To keep the manuscript focused and concise we included only figures showing quantitative data in the main text of the original manuscript. Since the other two receivers did not ask for any supplementary material to be moved to the main text we made a compromise. We agree that the figures showing phenotype and morphology examples will help to illustrate the biocontrol and safety aspects of the E. nigru B14-1 and should be in the main text. The rest were left in the supplementary.
The following figures have been moved to main text:
- mycelium and spore morphology (previously supp fig 1) is now updated with B14-1 colony morphology on PDA and moved to main text as figure1
- examples of biocontrol phenotypes (previously supp fig 4) in now figure 2
- result of fruit inoculation with B14-1 (previously supp fig 5) is now fig 5
To keep the manuscript focused and concise we left the following figures in the supplementary:
- Phylogenetic trees (previously supp figure 2,3) are now supp. figure 1 and 2
- Representative plant phenotypes observed on B14-1 and control plants are have been added as Supp. figure 3
- rootstock diameter profiles (previously Supp. figure 5) are now Supp. figure 4
- In-vitro biocontrol of Phytophthora cactorum (previously Supp figure 7) is now supp figure 5.
Reviewer 2 Report
Typing error at various places: "(Error! Reference source not found..)", I think it refers to supplementary materials, please correct it.
Suggestion: The abstract is well-written and organized. However quantitative results can be added in abstracts for the benefit of readers.
Author Response
Response:
Thank you for your inputs and recommendations. We have amended the manuscript according to your suggestions.
Typing error at various places: "(Error! Reference source not found..)", I think it refers to supplementary materials, please correct it.
Response: References to the supplementary materials updated.
Suggestion: The abstract is well-written and organized. However quantitative results can be added in abstracts for the benefit of readers.
Response: Quantitative data added to the abstract as suggested.
Reviewer 3 Report
To the authors,
The proposed manuscript about a potential BCO against apple canker handles a broad topic of the efficacy of potential BCO. As the authors mentioned most promising candidates fail in field tests. As such, it was delightful to read to read this manuscript with successful field tests.
Overal the manuscript is logically and well written leading to only some minor comments and remarks.
Overall, there is an error with the links to your supplemental data (line 323,326,332,402
Line 220-221: It is not clear for me if these apples were collected from the plots of the previous field antagonism assays or not. Can you clarify this?
Line 279: In this section, you never mention if it is a foliar application or spraying the roots instead of drenching the roots. Yet, from the supplementary data, you indeed mention in the caption that you mean foliar application. So, add this in this section to be sure.
Line 353: You say that canker incidence was reduced by 5.3%. but when I made the calculation (as i also did for line 341), my result is 5.5% ?
Figure 2: You should switch the labels putting control first followed by B14-1 treatment. As such, it is similar as it is presented in figure 1.
Line 445: More a remark or suggestion but shouldn't it be interesting to repeat it on other commercially apple cultivars both susceptible and less-susceptible cultivars? As you already highlighted in Line 65, there was a difference in overall fungal community between these cultivars leading to different susceptibility towards cancer. As such, can it be possible that on less-susceptible cultivars, E. nigrum abundance after 1 year of inoculation will be higher than on susceptible cultivars as you showed in figure 2? (I assume Royal Gala is susceptible to apple cancer?)
Yours sincerely
Author Response
The proposed manuscript about a potential BCO against apple canker handles a broad topic of the efficacy of potential BCO. As the authors mentioned most promising candidates fail in field tests. As such, it was delightful to read to read this manuscript with successful field tests.
Overall the manuscript is logically and well written leading to only some minor comments and remarks.
Response: We thank the reviewer for kind and reassuring comments and corrections suggested.
- Overall, there is an error with the links to your supplemental data (line 323,326,332,402
Response: References to the supplementary materials have been updated.
- Line 220-221: It is not clear for me if these apples were collected from the plots of the previous field antagonism assays or not. Can you clarify this?
Response:
Thank you for pointing this out. For quantification of B14-1 at 10 and 20 dpi we sampled trees from a specifically dedicated experiment/plots.
For quantification of B14-1 one year after inoculation we sampled tissues that were inoculated in the previous year as a part of the antagonism trials.
This was originally described in a paragraph that was ca. half a page below the line highlighted by the reviewer. The previous paragraph order did not follow the logical sequence and has been amended. We have moved the text describing the sampling/experimental procedures above the text describing quantification of B14-1 in plant tissue. We have also changed the paragraph subtitle (line 223) to clearly denote that it considers sampling and experimental design and pointed out the 2 experiments clearly in the text (line 224 and 241). We hope that this is now clear.
- Line 279: In this section, you never mention if it is a foliar application or spraying the roots instead of drenching the roots. Yet, from the supplementary data, you indeed mention in the caption that you mean foliar application. So, add this in this section to be sure.
Response: Information added as requested to lines 285-286.
- Line 353: You say that canker incidence was reduced by 5.3%. but when I made the calculation (as i also did for line 341), my result is 5.5% ?
Response: We would like to thank the reviewer for pointing out the error. We have double checked the figures and recalculated the % reduction. The text has been amended with correct numbers in lines 364, 375-376.
- Figure 2: You should switch the labels putting control first followed by B14-1 treatment. As such, it is similar as it is presented in figure 1.
Response: Figure label order amended as suggested. This figure in now figure 4 in the updated manuscript.
- Line 445: More a remark or suggestion but shouldn't it be interesting to repeat it on other commercially apple cultivars both susceptible and less-susceptible cultivars? As you already highlighted in Line 65, there was a difference in overall fungal community between these cultivars leading to different susceptibility towards cancer. As such, can it be possible that on less-susceptible cultivars, E. nigrum abundance after 1 year of inoculation will be higher than on susceptible cultivars as you showed in figure 2? (I assume Royal Gala is susceptible to apple cancer?)
Response: Thank you for another good point. We added this to the discussion section in lines 503-507.